# Synthesis of Poly-Sodium-Acrylate (PSA)-Coated Magnetic Nanoparticles for Use in Forward Osmosis Draw Solutions

**DOI:** 10.3390/nano9091238

**Published:** 2019-08-31

**Authors:** Irena Ban, Sabina Markuš, Sašo Gyergyek, Miha Drofenik, Jasmina Korenak, Claus Helix-Nielsen, Irena Petrinić

**Affiliations:** 1Faculty of Chemistry and Chemical Engineering, University of Maribor, Smetanova ulica 17, SI-2000 Maribor, Slovenia; 2Department for Materials Synthesis, Jožef Stefan Institute, SI-1000 Ljubljana, Slovenia; 3Department of Environmental Engineering, Technical University of Denmark, Miljøvej 113, 2800 Kgs. Lyngby, Denmark

**Keywords:** magnetic nanoparticles, poly-sodium-acrylate, osmotic pressure, forward osmosis, draw solution

## Abstract

The synthesis of magnetic nanoparticles (MNPs) coated with hydrophilic poly-sodium-acrylate (PSA) ligands was studied to assess PSA-MNP complexes as draw solution (DS) solutes in forward osmosis (FO). For MNP-based DS, the surface modification and the size of the MNPs are two crucial factors to achieve a high osmolality. Superparamagnetic nanoparticles (NP) with functional groups attached may represent the ideal DS where chemical modifications of the NPs can be used in optimizing the DS osmolality and the magnetic properties allows for efficient recovery (DS re-concentration) using an external magnetic field. In this study MNPs with diameters of 4 nm have been prepared by controlled chemical co-precipitation of magnetite phase from aqueous solutions containing suitable salts of Fe^2+^ and Fe^3+^ under inert atmosphere and a pure magnetite phase could be verified by X-ray diffraction. Magnetic colloid suspensions containing PSA-coated MNPs with three different molar ratios of PSA:MNP = 1:1, 1:2 and 1:3 were prepared and assessed in terms of osmotic pressure, aggregation propensity and magnetization. Fourier Transform Infrared Spectroscopy (FTIR) confirmed the presence of PSA on coated MNPs and pristine PSA-MNPs with a molar ratio PSA:MNP = 1:1 exhibited an osmotic pressure of 30 bar. Molar ratios of PSA:MNP = 1:2 and 1:3 lead to the formation of less stabile magnetic colloid solutions, which led to the formation of aggregates with larger average hydrodynamic sizes and modest osmotic pressures (5.5 bar and 0.2 bar, respectively). After purification with ultrafiltration, the 1:1 nanoparticles exhibited an osmotic pressure of 9 bar with no aggregation and a sufficient magnetization of 25 emu/g to allow for DS regeneration using an external magnetic field. However, it was observed that the amount of PSA molecules attached to the MNPs decreased during DS recycling steps, leaving only strong chelate-bonded core-shell PSA as coating on the MNPs. This demonstrates the crucial role of MNP coating robustness in designing an efficient MNP-based DS for FO.

## 1. Introduction 

Forward osmosis (FO) processes based on membrane technologies have garnered increased attention due to the great potential for lower energy techniques in wastewater treatment and desalination compared with a conventional reverse osmosis (RO) process [1]. The benefits of such a process, apart from the pressure reduction that operates on the membrane, come from the reduction of the energetic cost of the water permeation through the membrane [2]. Here, water molecules in a feed solution (FS) with low osmotic pressure are drawn across a semipermeable membrane to a so-called draw solution (DS) with high osmotic pressure [3]. In addition, it must exhibit minimum reverse transport from the DS side to the feed side, be easily separated and re-used upon water extraction or be readily available if regeneration is not required. 

The major benefits of FO versus RO include a high amount of water recovery, lower energy use and lower propensity for membrane fouling. However, a major challenge in FO, when used to produce clean water is how to re-concentrate the DS. Thus, the draw solutes should allow for reuse in the FO process, and retain its high performance in the long-term operation. Many studies have been performed to identify appropriate draw solutes over the past few decades [4]. Based on the available literature, NaCl appears to be the most promising DS (approximately 40% of experiments), due to its high solubility, low cost and relatively high osmotic potential. It has been used as a DS in concentrations from 0.3 M to 6 M, but is often used at 0.5 M, simulating the osmotic pressure of seawater and prompting the use of real seawater or RO brine as a DS [5]. Nevertheless, the type of wastewater (feed solution) and the required product purity have influence on the DS selection. Some studies report that using magnetic and/or hydrophilic nanoparticles as a DS [6,7] could be ideal candidates given their recovery facility from the diluted draw solution by means of a magnetic field as well as they eliminate the possibility of reverse draw solute flux [8]. 

Aqueous dispersions of bare magnetite nanoparticles have negligible osmotic pressure, even at the maximum concentration attainable in any FO plant roughly (50 %, wt %) [7] and have an isoelectric point close to neutral pH, therefore, they are not stable and can easily aggregate in aqueous solutions [8]. Therefore, magnetic nanoparticles (MNPs) need to be functionalized with molecules that generate osmotic pressure and lower the surface reactivity of the MNPs. It is reported that polyelectrolyte-coated MNPs featured an osmotic pressure many folds higher than free polyelectrolyte [9]. The unique properties of these hybrid organic-inorganic MNPs fulfil the requirements desired for an efficient DS: high surface area-to-volume ratio, versatile surface modification to modify the osmotic properties, superparamagnetic properties to ease their recovery and recycling, nontoxicity and large molecular size for better prevention of back solute flux [10]. Various coating polyelectrolyte were reported in the literature [11] and the choice of the coating polyelectrolyte for its application of MNPs as DS in FO is still poorly understood. Other synthetic materials were also investigated as draw solutes. Some stimuli-responsive polymer hydrogels have been developed as draw solutes [12]. Hydrogels are three-dimensional networks of polymer chains that are cross-linked by either physical or chemical bonds and are able to be understood to trap large volumes of water attracted by the high concentration of hydrophilic groups. Similar to other organic draw agents, stimuli-responsive polymer hydrogels also benefit from a negligible reverse flux due to their macromolecular structure [13]. However, polymers without ionic groups generally have lower osmotic pressure than those with ionic groups. It was found that the overall FO flux using pure cross-linked poly N-isopropylacrylamide (P-NIPAAm), which is a classical termo-sensitive hydrogel composed of both hydrophilic and hydrophobic groups, is still very low (<0.1 LMH, L/m^2^h) [14] compared with the inorganic draw solute (typically ~20 LMH), L/m^2^h [15].

In this paper, we investigate the synthesis and characterisation of MNPs coated with hydrophilic poly-sodium acrylate (PSA), where the degree of ionization depends on the solution pH. The hydrophilic -COOH groups adsorb onto the iron-oxide surface by forming a chemical bond of the chelate type. PSA-MNPs have already been reported to be draw solutes in FO processes [7,8,16,17]. Dey and Izake reported that the PSA conformation (extended or coiled) significantly affects the osmotic pressure and depends on the PSA concentration attached to the surface. An averaged water flux of 5.3 L/m^2^h and an osmotic pressure of 11.4 bar were obtained when using PSA-MNPs at very low DS concentration (0.078 %, wt %, ~ 1.3 g/L), where the polyelectrolyte chain remains extended. Ge and co-workers [16] reported an osmotic pressure of 11–12 bar with a concentration of (~15 %, wt %) with PSA (Mn 1800) as the draw solute. Others [7,17] report the use of (24–48 %, wt %) PSA draw solute (in the form of free polyelectrolyte osmotic agent) to generate the similar osmotic pressures (~11 bar). 

The main issue in using PSA-MNPs as DS is related to the robustness of the PSA coating, since the PSA-MNP interaction involves a substantial amount of non-chelate-type interactions, these may give newly-synthesized pristine PSA-MNPs with high PSA loading and high osmotic activity but the non-chelate PSAs may release from the coated MNP during the recycling steps involved in DS recycling which diminishes the osmotic pressure after repeated DS recycling. 

In order to examine the stability of the PSA-coated MNPs and their technological applicability as DS, we investigate the synthesis and characterisation of MNPs coated with PSA where three different molar ratios of PSA:MNP = 1:1, 1:2 and 1:3 were prepared by co-precipitation of Fe^3+^ and Fe^2+^ sulfates salts with hydrophilic PSA using ammonia in a one pot synthesis and purified with ultrafiltration (UF). PSA-MNPs were characterized and assessed in FO using biomimetic aquaporin (AqP) flat sheet membrane.

## 2. Materials and Methods 

### 2.1. Chemicals 

All chemicals were of reagent grade and used without further purification. Iron(II) sulfate heptahydrate (FeSO_4_∙7H_2_O > 99.5%) was obtained from Acros organics, Geel, Belgium. Iron(III) sulfate x hydrate (Fe_2_(SO_4_)_3_∙xH_2_O, 21%–23%) were purchased from Sigma–Aldrich, St. Louis, Missouri, USA. Polyacrylic acid (PSA) (sodium salt solution, 1200 g/mol, (45 %, wt %) in H_2_O, ethanol (C_2_H_5_OH > 99%) and ammonium (NH_3_) (25% NH_3_ in H_2_0) were all obtained from Merck Co, Kenilworth, N. J., USA. 

### 2.2. Preparation of PSA-MNPs Solutions

Aqueous solutions of MNPs were prepared by alkalinizing an aqueous mixture of ferric and ferrous salts with ammonia at room temperature [18]. Stock solutions of MNPs:0.13 M FeSO_4_×7H_2_O, 0.05 M Fe_2_(SO_4_)_3_×H_2_O in 10 mL deionized water (DI) was prepared as a source of iron by dissolving the respective chemicals in DI water under vigorous stirring (600 rpm/min) for 10 min and PSA was added (appropriated quantity of PSA was dissolved in 10 mL DI water). The formed solution was heated to 80 °C with the addition of 20 mL NH_3_ (25% water solution NH_3_). The color of the suspension turned to black almost immediately. During the synthesis, the content of the flask was bubbled with N_2_.


**Stock solution PSA:**


Two mmol PSA was dissolved in 10 mL DI water (the calculation of the amount of PSA was based on the number of COONa groups in polymer).

Three different types of PSA-MNPs solutions were prepared according to the following procedures:Sample A: Stock solution PSA (20 mmol PSA in 10 mL DI water):Stock solution MNPs = 1:1,Sample B: Stock solution PSA (1 mmol PSA in 10 mL DI water):Stock solution MNPs = 1:2 andSample C: Stock solution (0.66 mmol PSA in 10 mL DI water) PSA:Stock solution MNPs = 1:3.

All three solutions were stirred for a further 30 min at 80 °C. The solutions were subsequently allowed to cool to room temperature. The PSA-MNPs were first isolated using a permanent magnet (0.8 T). However, the particles were very stable, and it was not possible to separate them using the magnet. Therefore, ultrafiltration (UF) was used and the samples were filtrated in a dead-end cell (Millipore, stirred cell, USA, UF membrane was Nadir UP 020, MWCO = 20,000 g/mol) for 1 h. After separation, samples were rinsed (also in UF cell) with a 50 mL mixture prepared of ethanol/DI water = 1:1 (molar ratio) and additionally with 50 mL of DI water. After rinsing and ultrafiltration at the final concentration of 0.1 g/L, the sample was sonicated in an ultrasound bath (Iskra Sonis 10, Šentjernej, Slovenia)) for 1 h. The final pH of the MNPs suspension was 6.5.

### 2.3. Characterization of Nanoparticles 

#### 2.3.1. Structural X-Ray Powder Diffraction

Dried product was characterized by X-ray Diffraction (XRD) patterns using Siemens D5005 diffractometer with a monochromatic in the diffracted beam (Bruker, Hamburg, Germany). From the XRD peak broadening and Scherrer equation (Equation (1)), the mean size of the ordered (crystalline) domains (average nanoparticle size) (*τ*) was estimated [19]:(1)$$\tau =\frac{K\lambda }{\beta \cos\theta }$$
where, *K* is the dimensionless shape factor, with a value close to unity (0.9), *λ* is the X-ray wavelength (m), *β* is the line broadening at half the maximum intensity after subtracting the instrumental line broadening (rad), and *θ* is the Bragg angle (°).

#### 2.3.2. Fourier Transform Infrared Spectroscopy (FTIR)

The functional groups of PSA-MNPs were investigated by Perkin-Elmer 5000 Fourier Transform Infrared Spectroscopy (FTIR) spectrometer (PerkinElmer Inc, Beaconsfiled, UK). The pure surface modification agent, i.e., PSA, was also investigated for comparison.

#### 2.3.3. Surface Charge, Hydrodynamic Diameter, and Size Distribution Measurements 

To determine the surface charges of the PSA-MNPs, dynamic light scattering (DLS) analysis was used. The purified PSA-MNPs water dispersion of sample A was diluted to 0.1 g/L with DI water. The pH values were adjusted from 3 to 11 by an autotitrator during measurements and the zeta potentials (ζ-potential), particle size and hydrodynamic size distribution of modified Fe_3_O_4_ MNPs dispersed in water were measured using Zetasizer Nano-S (Malvern Instruments, Malvern, UK), by application of the Henry equation to obtain the Zeta potential of the particles [20]. The refractive index used for size and zeta potential measurements was 1.418 (PSA).

#### 2.3.4. Particle Morphology Determined by Transmission Electron Microscopy (TEM)

The particles were observed by transmission electron microscopy (TEM) using Jeol JEM-2100 (JEOL, Peabody, Massachusetts, USA, Inc.), where the nanoparticles were deposited on a copper-grid-supported, perforated, transparent carbon film. The surface ligand concentration *n_s_* (the number of ligands per particle) were estimated using Equation (2) [21]:(2)$$n_{s}=\omega \times 1/6\pi d^{3}\rho_{Fe_{2}O_{3}}\times 6.023/M_{PSA}\left(1-\frac{\omega }{100}\right),$$
where, *ω* is weight loss (%), *d* is particle diameter (nm), estimated from the TEM images to be 4 nm assuming spherical-shaped nanoparticles, *ρ _Fe2O3_* is 4.86 g/cm^3^ and *M_PSA_* is 1200 g/mol. 

#### 2.3.5. Characterisation by Thermogravimetric Analysis

Thermogravimetric analysis (TGA) was performed on the dried powdered samples (5 mg) with a heating rate of 20 K/min using a Mettler TGA thermogravimetric analyzer TGA/SDTA, 851, (Mettler Toledo, Switzerland) in air and N_2_ atmosphere up to 900 °C. Mass spectroscopy was measured with (Balzers Quadstar 422 V 5.0, Balzers, Liechtenstein) in nitrogen and air atmosphere up to 700 °C. Changes in weight were ascribed to changes in the polymer coating layer PSA on the PSA-MNPs nanocomposite. 

#### 2.3.6. Osmolality Measurements

The osmolality of the solutions (∑φ·n·c) was determined via freezing-point depression (Gonotec, cryoscopic osmometer – OSMOMAT 030, Berlin, Germany) and the osmotic pressure, π of PSA-MNPs solutions was calculated using Equation (3) [22]:(3)π =(∑φ·n·c)·R·T,
where, *R* is gas constant (8.31447 kJ/mol K), *T* is absolute temperature (K), *N* is amount of substance (mol), *C* is molarity of a solution (mol/L) and *φ* is osmotic coefficient (-).

#### 2.3.7. Magnetization Study by Vibrating Sample Magnetometry 

The saturation magnetizations of PSA-MNPs were characterized using a vibrating sample magnetometer (VSM, Lake Shore 7307, Westerville, Ohio, USA). For sample preparation, the nanocomposite powder (dried at 353 K for 12 h) was placed in a plastic holder and was sealed. Measurements were conducted at 300 K with an external magnetic field ranging from −10 × 10^3^ to 10 × 10^3^ Oersted (Oe) using a magnetic separator (Model L-1, S. G. Frantz Co., Tullytown, PA, USA).

### 2.4. Forward Osmosis (FO) Experiments

A cross-flow membrane cell (Sterlitech Corporation, Kent, Washington, USA) was fitted with a flat sheet FO AIM™ membrane (Aquaporin A/S, Kgs. Lyngby, Denmark). Batch experiments were conducted with the membrane in FO mode (active layer facing the feed side where the effective membrane area was 33.15 cm^2^ (Figure 1). The FS and DS were circulated (50 mL/min) by peristaltic pumps. The FS tank was positioned on a digital balance (Ohaus Scout Pro, Shanghai, China) connected to a computer and weight changes were recorded automatically every 30 seconds to determine the permeate water flux (*J_w_*). In addition, the conductivity of FS was monitored by a conductivity meter (Lovibond SD 320 Con, Dortmund, Germany) for the calculation of reverse solute flux (*J_s_*).

DI water was used as a FS and PSA-MNPs (7 %, wt %) were tested as a DS. FO experiments were performed at room temperature 23 °C ± 0.5 °C. Water flux through the membrane was decreasing over time as the DS was diluted, thereby lowering the osmotic driving force across the membrane, and experiments were stopped manually when *J_w_* decreased to 50% of the initial value.

Water flux, *J_w_* (L/m^2^h), across the FO membrane was calculated based on volume change of FS using Equation (4) [23]:(4)$$J_{w}=\frac{\Delta V}{A\;\Delta t}$$
where, *ΔV* is the total volume change of permeate water (L), *A* is the effective membrane area (m^2^) and *Δt* is time (h).

In FS, conductivities were measured and converted to concentrations (in mg/L) using a standard curve built from a series of single solutions relating conductivity and concentration. The concentration of the DS transporting to the FS was thereafter obtained directly from the standard curve. The reverse solutes flux, *J_s_* (g/m^2^h), was determined from the concentration increase of the FS using Equation (5) [24]:(5)$$J_{s}=\frac{c_{t}V_{t}{-c}_{0}V_{0}}{A\;\Delta t}$$
where, *c_0_* is the initial concentration of the feed solution (g/L), *V_0_* is the initial volume of the feed solution (L), *c_t_* is the solute’s concentration (g/L), *V_t_* is the volume of the feed solution measured at time (L) and *Δt* is time (h). 

## 3. Results and Discussion

### 3.1. Analyses of Uncoated and Coated Samples A, B and C Using XRD, FTIR, Thermogravimetric Analysis (TGA) and Zeta Potential 

MNPs were prepared by the co-precipitation technique from solutions of Fe^2+^ and Fe^3+^ ions in molar ratio of 1:2 with ammonia at room temperature (Equation (6)): (6)$${\mathrm{Fe}}^{2+}+2{\mathrm{Fe}}^{3+}+8{\mathrm{OH}}^{-}\to {\mathrm{Fe}}_{3}O_{4}+4H_{2}O,$$

To lower the surface reactivity due to the large ratio of surface area-to-volume, MNPs tend to agglomerate in order to reduce their surface energy by strong magnetic dipole-dipole attraction between particles. To mitigate this, hydrophilic PSA is used as a stabilizing agent against aggregation and flocculation and added after the addition of the ammonia base, thus allowing particles to grow before coating. X-ray diffraction (XRD) analyses confirmed the crystalline structure of as-prepared magnetic nanoparticles (Figure 2).

The three most intense peaks (311, 440 and 511) correspond well to values previously reported for magnetite (Fe_3_O_4_) crystals [25]. 

The average size of magnetite nanoparticles was estimated, using Equation (1), to be 4 nm and confirmed with TEM analysis (Figure 3).

The success in synthesizing the PSA-MNPs was confirmed by FTIR spectroscopy. The chemical composition of the 45% water solution of PSA and coated MNP particles (sample A) was examined by FTIR and is presented in Figure 4. 

PSA displays characteristic peaks at 1400 cm^−1^, 1451 cm^−1^ (COO- symmetric stretch), and 1542 cm^−1^ (COO- asymmetric stretch) respectively, indicating the presence of PSA on the coated MNPs. The broad band between 520 cm^−1^ and 580 cm^−1^ is characteristic of Fe-O vibrations in iron oxide [9].


**TGA and Mass Spectroscopy**


In order to determine the amount of PSA grafted to the surface on nanoparticles, thermogravimetric analysis was used. The poly-acrylates degradation mechanism includes dehydration, the formation of anhydride-type structures and their chain scission and de-polymerization. Above 340 °C, pyrolysis in air yields thermo-oxidation and complete decomposition while pyrolysis in argon yields decarboxylation and a carbonaceous residue [26]. 

Overall, the weight loss is the loss of the organic part, reduced by the loss of adsorbed water up to 150 °C. The final mass loss is related to the final amount of PSA attached on the MNPs compared to the starting composition ratio.

Colloid solutions of PSA-MNPs were separated by UF and dried for further thermogravimetric analysis. The weight loss of samples A, B and C in air atmosphere is presented in Figure 5.

The estimated mass loss for sample A, sample B, and sample C was 37%, 22%, and 15%, respectively. Differences in mass loss corresponds to different molar ratio composition, namely PSA:MNP for each sample. 

The estimated surface ligand concentration was 0.95, 0.46 and 0.29 molecules per nm^2^ for sample A, sample B and sample C, respectively (Table 1). The estimated surface ligand concentrations are believable and did not define the status of the PSA ligands, i.e., which (part) of ligands is bound to the surface by chelate bounds and which (part) with physical bounds. 

In order to gain more insight into the PSA attachment to MNPs, analysis of oxidation decomposition heat curves and residual mass loss under different conditions (air and N_2_) were conducted, see Figure 6. Thermo-gravimetric analysis (TGA) was conducted on the dried powdered sample (5 mg) with a heating rate of 20 °C/min in air and N_2_ up to 900 °C. By heating in N_2_, two degradation steps are seen, whereas in air this is not the case. Thus, weight loss in N_2_ occurs with an initial degradation beginning at ambient temperature and ending at about 450 °C, the second step above 450 °C starts with a plateau which extends to 650 °C where a notable weight loss occurred associated with decarboxylation and CO_2_ (or/and CO) release. At the end (mass loss after 600 °C), the carbonaceous residue reacts with iron oxide forming wustite (FeO) (the CO_2_ is additionally released). 

This behavior is also observed in the TGA and mass spectra of sample A obtained in a nitrogen atmosphere and air, see Figure 7. In air, a gradual weight loss continuously proceeded up to 700 °C with a strong expulsion of CO_2_ and H_2_O (other mass fragments were not followed). The final product was hematite (α-Fe_2_O_3_), as identified by XRD spectral analysis.

The mass loss and spectroscopy analyses clearly showed the delay of the disintegration in nitrogen which is consistent with the different atmospheric conditions and due to the tightly bonded PSA to the iron oxide which reacted only at higher temperatures. The surface ligand weight loss in sample A is relatively high. The radius of gyration *R*_g_ of a polyacrylic acid in aqueous media, *M_w_* = 1200 g/mol, and degree of polymerization around 12 is 0.7 nm [27]. This translates into a tight planar surface packing of around 0.64 PSA molecules/nm^2^ on the MNP. This estimation is rough, because PSA coils may significantly shrink due to higher ionic strength (Na^+^ ions) in comparison with polyacrylic acid and curved surfaces can accommodate more molecules per surface area than planar. In any case, the weight loss-determined surface ligand concertation (Table 1) is significantly higher as the estimated maximum surface concentration, indicating more than one type of PSA association to MNP.

Taken together, the results suggest that the PSA ligands are attached to MNPs in two modes i.e., i) a layer where the molecules are attached to the particle surface with strong chelate bonds and ii) an outer layer where the ligands are bonded with physical bonds in-betweens PSA molecules as well as to the first layer (Figure 8). Thus, when the amount of PSA is larger than needed to form a single molecular layer, the surplus is attached non-covalently to the PSA-MNP complex.


**Zeta Potential Determination of Colloidal Suspensions**


The colloidal properties of the suspensions were examined by measuring the *ζ*-potential and are presented in Figure 9 for sample A. The *ζ*-potential of sample A was measured as a function of the suspension pH. The starting pH for the measurement was the intrinsic pH (7.8) of the diluted suspension. The ammonia was titrated into the suspension to measure the *ζ*-potential at higher pH values (pH > 7.8). The absolute value of the *ζ*-potential is linked to the electrostatic repulsive forces that avoid aggregation of the PSA-MNPs in aqueous suspensions. Generally, it was recognized that stable suspensions with PSA or poly(acrylic acid) PAA appear if the *ζ*-potential exceeds a value of ±30 mV [9,28].

An effective way to promote the water stability of MNPs is to change the isoelectric point by surface modification. The *ζ*-potential becomes more negative with increasing pH due to the deprotonation of the carboxylate groups of PSA. The dispersed particles showed stability up to a pH value of 7. At a lower pH, the flocculation started and the PSA-MNPs sediment in the form of large agglomerates. The zeta potential of the other two samples, B and C, were measured only at pH 7.5 and the results are given in Table 1. 

Characteristics of the samples A, B, and C are pH, ζ-potential at pH 7.5 osmotic pressure, weight loss and ligand surface concentration, as presented in Table 1. 

Sample A was taken for further analyses based on osmotic pressure data and TGA analyses (sample A is the sample with highest osmotic pressure and amount of PSA). 

### 3.2. Analyses of Sample A


**Purification of Sample A**


In general, dialysis may be used to separate nanoparticles from the solutions in which they were formed where the eluate consists of non-complexed reactants and components. Sample A underwent a series of UF-based dialysis steps, see Table 2. The decrease of the osmotic pressure of sample A after the purification steps is likely caused by a decrease of PSA ligands complexed to the MNP which in turn leads to increased aggregation. When particles are agglomerated, the chemical activity of deprotonated COO^-^ groups is lower, and consequently, the osmotic pressure of the MNP solution is lower. This is substantiated by the observation that sonication of PSA-MNPs suspensions after the second and third purification step could partially recover osmotic pressure. De-aggregation was further confirmed by hydrodynamic size distributions of PSA-MNPs. 


**Size Distribution**


The hydrodynamic size distribution of PSA-MNPs was measured with dynamic light scattering DLS. The DLS measurements indicate no aggregation occurring after the second and third purification step. The polydispersity index (PDI) decreases with the number of purifications. After sonication, the average size of particles is smaller, as shown in Figure 10. 

The results demonstrate that the one pot synthesis offers both advantages and disadvantages. Clearly, it can produce small sized superparamagnetic particles stabilized by ligand attachments and these particles can generate significant osmotic pressures. On the other hand, the particles display two types of PSA attachment to the MNP, a tight-bound and a loose-bound type. For the latter, the PSA molecules are likely bonded with molecular forces to each other and to the tight-bound PSA layer.

During the synthesis of PSA-MNPs-based DS, the specific surface area and/or the number of surface iron atoms must be tuned with the initial PSA amount to achieve a single PSA covering layer. In this case, only a monolayer of tight-bound PSA molecules will govern the enduring osmotic pressure. However, these nanocomposites might need additional steric stabilization to avoid agglomeration.

### 3.3. Evaluation of PSA-MNPs as Draw Solution (DS) in FO 

Batch experiments were performed to evaluate PSA-MNPs as DS in FO. In FO, water is driven by a transmembrane osmotic gradient from the FS being concentrated through a semi-permeable membrane into the DS being diluted. The overall feasibility of using PSA-MNP solutes in the DS can be evaluated by four parameters: 1) transport performance in terms of *J_w_*, which depends on the membrane water permeability and osmotic potential of the DS, 2) reverse solute flux *J_s_* from the DS side to the feed side which depends on the FO membrane’s ability to retain solutes on the DS side as well as the DS concentration, 3) robustness of the DS which depends on the physical-chemical stability of the solute, and 4) the recovery efficiency of the DS solute, which in this case is related to how well externally applied magnetic fields can re-concentrate the DS.

Here, deionized water was used as the feed solution and 7% solution PSA-MNP was used as a draw solute. Measurements were used in Equations (4) and (5) to determine average values of *J_w_* and *J_s_* and specific reverse salt flux was defined as *J_s_*/*J_w_* to provide a measure of the DS solute lost per litre of water produced during FO, as this ratio is related to the water–solute selectivity of the membrane. Evaluated average parameters from a 1 h FO process are presented in Table 3.

The *J_s_*/J_w_ value is low, which indicates a low permeation of PSA-MNPs into the feed solution. *J_w_* versus time measured during the FO process, using PSA-MNP as a DS, is presented in Figure 11. 

Water flux decreases stepwise during the FO process because of the DS dilution, and consequently, the driving force (difference in osmotic pressure) decreases. The conductivity in FS is increased during the experiment due to charged PSA-MNPs permeating from the DS to the feed solution combined with the volume decrease of FS. For comparison, *J_s_*/*J_w_* for AqP flat sheet FO membranes has been reported to be 0.1 g/L for NaCl and about 0.01 g/L for MgSO_4_ or MgCl_2_ [29]. 

After filtration, the FO membrane was taken out of the cell (Figure 12). A portion of nanoparticles stayed on the membrane (supported side) after cleaning. 

The integrity test with 1 M NaCl was performed to determine changes in *J_w_* on the membrane with accumulated nanoparticles on the support side of the flat sheet AqP membrane. After backwash, the second FO filtration with 1 M NaCl as a DS and deionised water as FS was performed. Obtained results are shown in Figure 13.

Even though the supported layer visually seemed clogged, it does not influence *J*_w_.

The ability to re-concentrate the DS depends on the magnetization of the functionalized particles. To have efficient extraction of PSA-MNPs from the water they must exhibit a suitable magnetization to ensure that no PSA-MNPs remain in the extracted clean water after applying an external magnetic field gradient. Figure 14a shows the magnetization-magnetic field (M-H) curve hysteresis of sample A, representing a typical M-H curve for superparamagnetic particles with radii around 3–4 nm. 

The room temperature magnetization of coated MNPs versus magnetic field exhibits a superparamagnetic behavior which can be seen from Figure 14a, where no sizeable hysteresis was observed verifying their superparamagnetic behavior at room temperature. Saturation was not achieved even at 1 T, because of the nanoparticles’ small size and consequently large surface-to-volume ratio. This is consistent with spin canting effects at the surface and a small MNP size. Samples showed a room temperature magnetization of 25 emu/g – pure magnetite exhibits about 60 emu/g [30]. 

This suggests the presence of about (41 %, wt %) of nonmagnetic phase, which is fairly close to the (37 %, wt %) of PSA estimated in sample A using TGA. The stabilized nanocomposites-coated MNPs (sample A) at a pH > 5 can be manipulated with a permanent magnet to remove the particles from the DS, as shown on Figure 14b.

Figure 14b suggests a viable separation mechanism based on an externally applied magnetic field, but applicability also critically depends on having a stable osmotic pressure of the colloidal solution. This depends on the concentration/number of PSA hydrophilic ligands and/or their hydrophilic functional groups. In the DS, PSA can be in three states: i) dissolved in the colloidal solution, ii) loosely bound to the particles in equilibrium with the solution concentration, and iii) bound to the particle surface with chelate bonds. Only the last state will determine the repeatable (re-cyclable) DS osmotic pressure. After the second purification step of sample A, the osmotic pressure decreases from 9 bar to 4 bar, and after the third purification step the osmotic pressures measured was 1.3 bar respectively, see Table 2. This is also reflected in UF filtrate conductivity (arising from free PSA monomers in solution) which decreases after successive UF dialysis steps, see Table 4. Thus, the applicability of PSA-MNPs investigated here is limited by the amount of stably-bound PSA as the loosely-bound PSA cannot be recovered using an externally applied magnetic field. We can also correlate the measurements of filtrate conductivity to the quantity of PSA monomers which are dissolved in the colloidal solution. This is the reason why the pressure drops after the second purification step. Ions close to the surface of the particle will be strongly bound while ions that are further away will be loosely bound, forming a diffuse layer. Within this layer, there is a notional boundary and any ions within this boundary will move with the particle when it moves in the liquid but any ions outside the boundary will stay where they are, in the slipping plane.

## 4. Conclusions 

The PSA-coated Fe_3_O_4_ nanoparticles have been successfully synthesized. Systematic analyses using zeta potential and surface-charge density were preformed to verify the hydrophilic nature of the PSA-coated MNPs obtained by co-precipitation. The thermogravimetric analysis reflects the composition ratios of MNPs/PSA. With increasing this ratio (syntheses A, B, C), the share of organic ligands capped to MNPs’ surface diminishes, and mass loss increases. 

The estimated surface ligand concentration includes all PSA ligands where one part is bound to the surface by covalent (chelate) bounds and another part is loosely associated to the PSA-MNP complex and the osmotic pressure of as-is PSA-MNP apparently was high enough to be used as a draw solute in FO. However, UF dialysis revealed a significant amount of loosely bonded PSA ligands, leaving only the firmly bonded ligand molecules on the particle surface, giving the DS its equilibrium properties. The rather modest osmotic pressure of the dialyzed DS limits the applicability of PSA-MNP as a solute molecule in its current state and additional steric stabilization will be needed for general use as DS solutes in FO. Therefore, further investigations will be conducted to establish a stronger (covalent) bond on loosely associated molecules. Some research using various (PSA and similar) molecules, bonded via carbodiimide activation of carboxylic groups, are still in progress.

## Figures and Tables

**Figure 1 nanomaterials-09-01238-f001:**
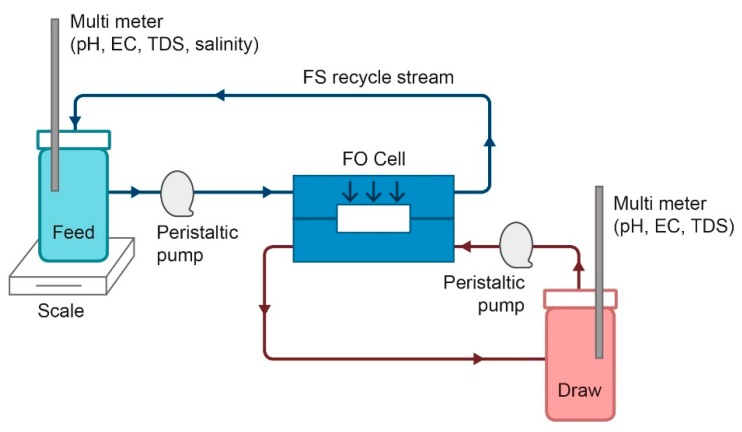
Process scheme of the laboratory-scale forward osmosis (FO) system.

**Figure 2 nanomaterials-09-01238-f002:**
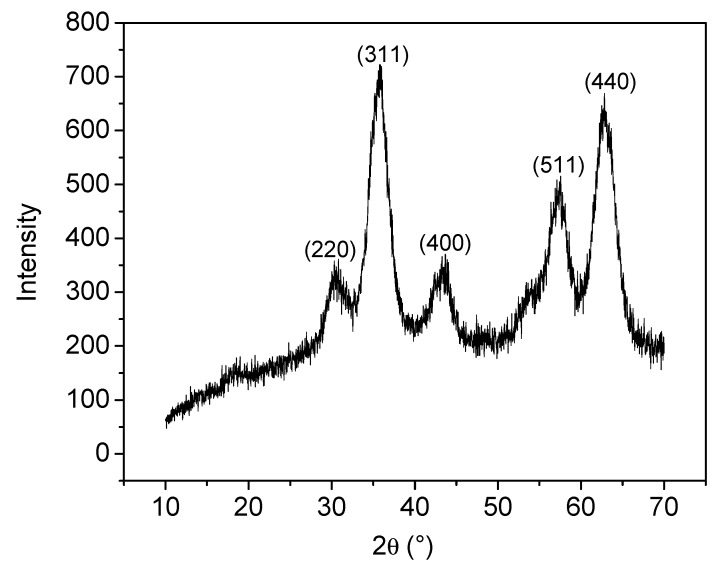
X-ray diffraction spectra of co-precipitated magnetic nanoparticles (MNPs) sample, prepared at room temperature. Miller indices indicated in brackets.

**Figure 3 nanomaterials-09-01238-f003:**
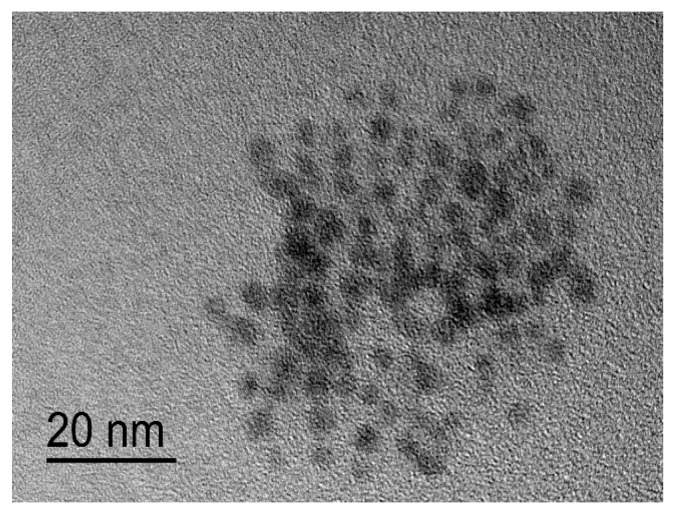
Transmission electron microscopy (TEM) of MNPs.

**Figure 4 nanomaterials-09-01238-f004:**
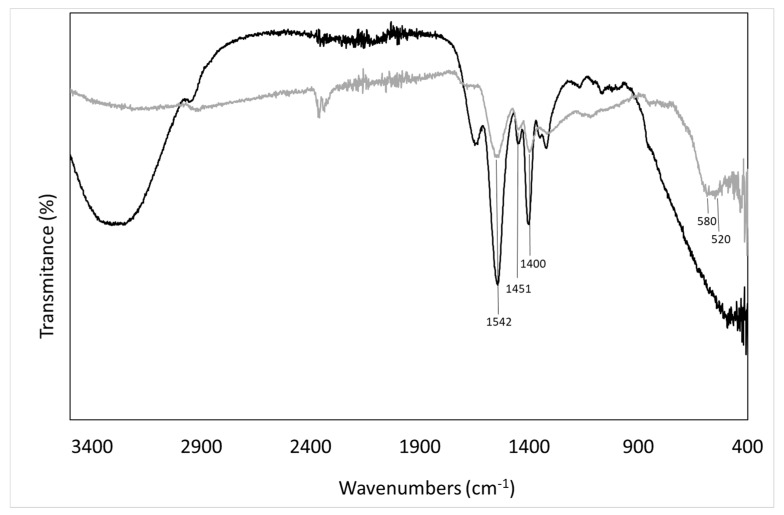
Fourier Transform Infrared Spectroscopy (FTIR) spectra of poly-sodium-acrylate (PSA): 45% water solution (black) and coated MNPs with PSA (gray).

**Figure 5 nanomaterials-09-01238-f005:**
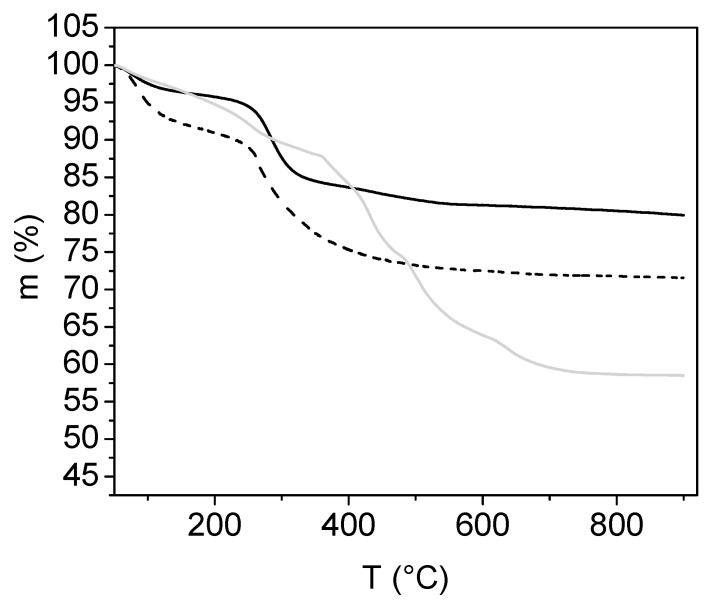
Mass loss of sample A (gray), sample B (dashed line), and sample C (black line), performed in air.

**Figure 6 nanomaterials-09-01238-f006:**
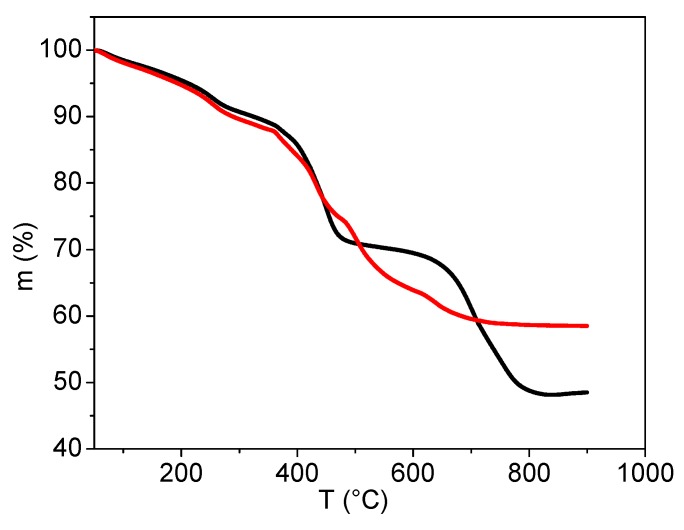
Mass loss of sample A performed in air (red) and nitrogen (black).

**Figure 7 nanomaterials-09-01238-f007:**
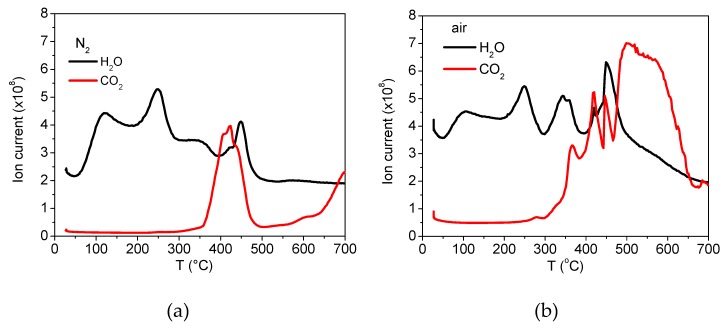
Mass spectra of PSA-MNPs determined by thermo-gravimetric analysis (TGA) in: (**a**) nitrogen and (**b**) air.

**Figure 8 nanomaterials-09-01238-f008:**
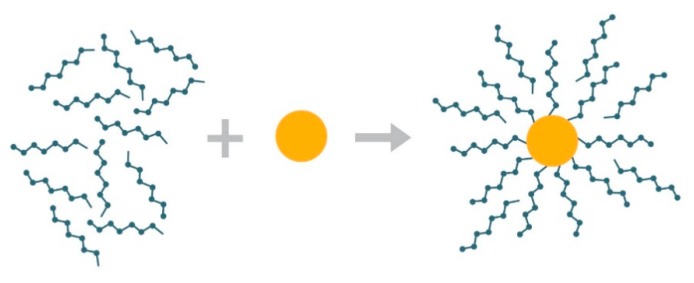
Proposed PSA-MNP assembly model: A layer of PSA monomers (green) is tightly bound to the MNP (orange) surrounded by loosely attached (here depicted as intercalating non-covalently bound) PSA molecules.

**Figure 9 nanomaterials-09-01238-f009:**
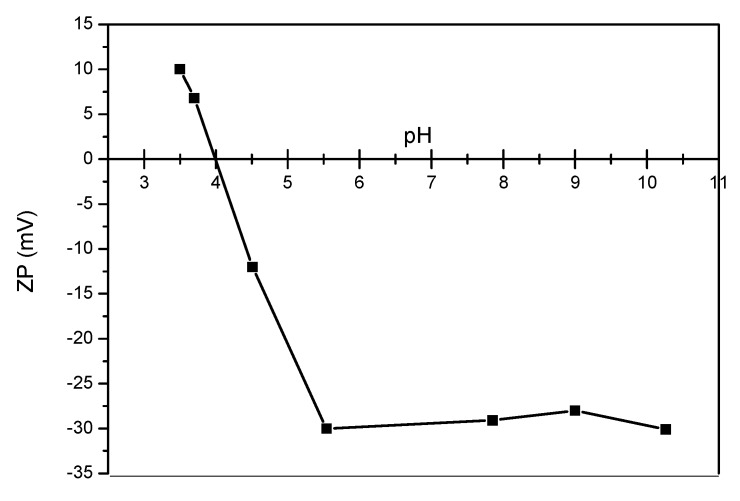
Effect of the solution pH on ζ-potential of MNPs for sample A. Increasing the pH decreases the positive charge as some of the carboxyl groups become negatively charged. At pH 4.1, the potential decreases to zero. Further increasing the pH beyond the isoelectric point increases the negative charge until pH 5.5 where the zeta potential reaches plateau.

**Figure 10 nanomaterials-09-01238-f010:**
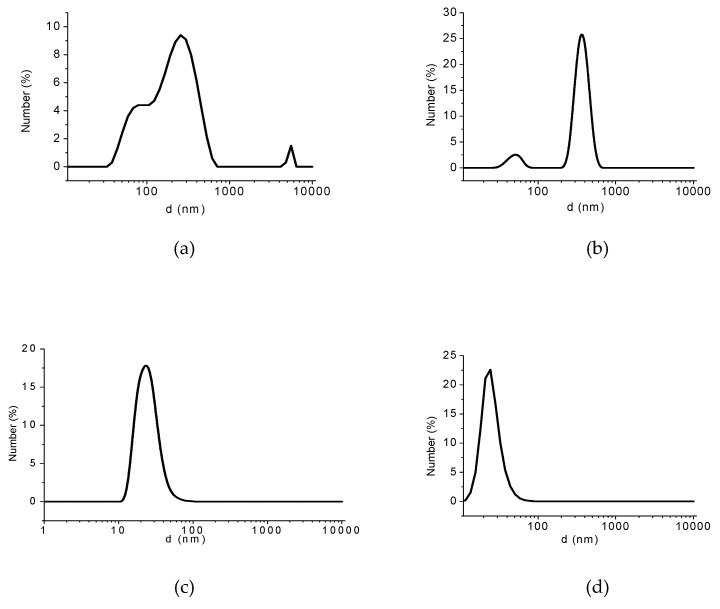
Average hydrodynamic size distribution of sample A: (**a**) as-prepared sample (polydispersity index (PDI) = 0.84), (**b**) after first UF purification step followed by sonication (PDI = 0.47), (**c**) after second UF purification step followed by sonication (PDI = 0.25), (**d**) after third UF purification step followed by sonication (PDI = 0.10).

**Figure 11 nanomaterials-09-01238-f011:**
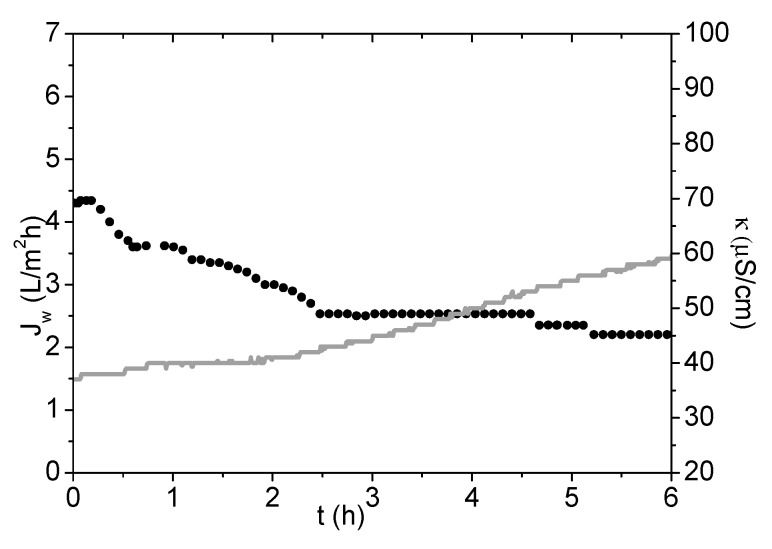
Water flux and feed solution (FS) conductivity measurements during FO process using PSA-MNP as a DS. Black dots presents *J_w_* values and grey line presents FS conductivity values referring to secondary Y-axis.

**Figure 12 nanomaterials-09-01238-f012:**
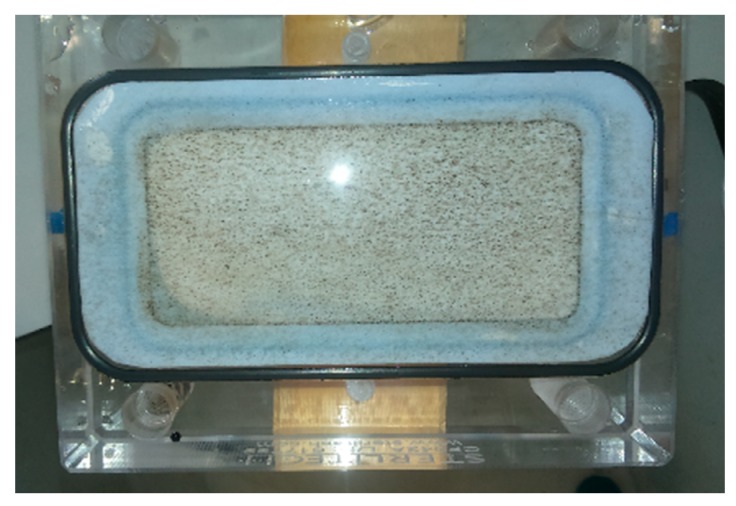
Photo of membrane support after using 7% solution MNP as a DS.

**Figure 13 nanomaterials-09-01238-f013:**
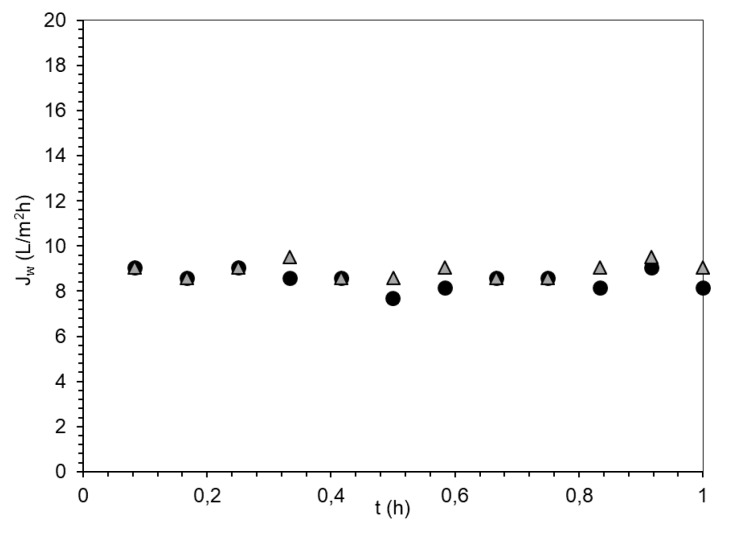
*J*_w_ versus time using 1 M NaCl and deionised water as a DS and FS on used membrane (
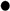
) and washed membrane (
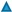
).

**Figure 14 nanomaterials-09-01238-f014:**
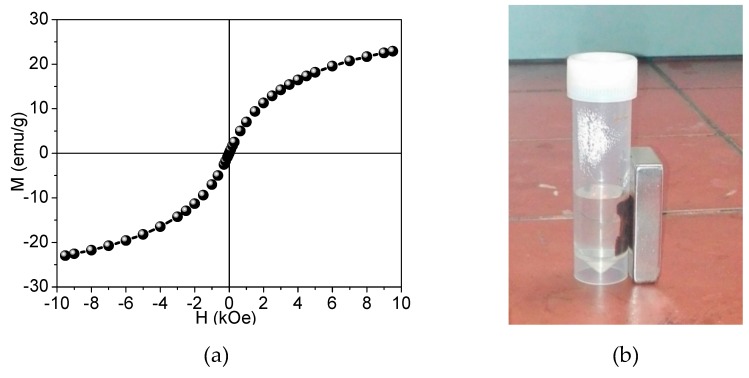
(**a**) Magnetization versus the magnetic field for sample A, (**b**) Separation of PSA-MNPs of sample A using a permanent magnet (0.8 T) on the flask wall.

**Table 1 nanomaterials-09-01238-t001:** Composition ratio, pH, ζ-potential, osmotic pressure of the suspensions, PSA-MNPs weight loss, ligand surface concentration of samples after first purification step (separation by ultrafiltration (UF)).

Molar Ratio PSA-MNPs.	pH	*ζ*-Potential (mV)	Osmotic Pressure (bar)	*m* (%) (air)	*n_s_* (molecules/nm^2^)
Sample A: 1:1	9.5	−42.9	9	37	0.95
Sample B: 1:2	8.3	−35.5	5.5	22	0.46
Sample C: 1:3	9.2	−41.8	0.2	15	0.29

**Table 2 nanomaterials-09-01238-t002:** Osmotic pressures and d values (hydrodynamic diameter) of successive purified samples.

Sample A.	Osmotic Pressure (bar)	Concentration (g/L)	d (nm)
As-prepared sample	30	0.1	520
Retentate from first UF purification	9	0.1	290
Retentate from second UF purification	4	0.1	300
Sonicated retentate after second UF purification	6	0.1	50
Retentate from third UF purification	1.3	0.1	630
Sonicated retentate after third UF purification	2.3	0.1	35

**Table 3 nanomaterials-09-01238-t003:** PSA-MNP Draw solutions used in FO process and average calculated parameters determined within 1 h running.

FS	DS	Initial Osmotic Pressure (bar)	J_w_ (L/m^2^h)	J_s_ (g/m^2^·h)	J_s_/J_w_ (g/L)	V_permeate in 1 h_ (mL)
DI water	PSA-MNP	9	3.8	0.05	0.01	10.6

where FS is feed solution, DS is draw solution, J_w_ is water flux, J_s_ is reverse solutes flux.

**Table 4 nanomaterials-09-01238-t004:** UF permeate conductivity of successively dialyzed samples.

Sample A	Conductivity (mS/cm)
Permeate from first UF purification	15.9
Permeate from second UF purification	2.64
Permeate from third UF purification	1.2

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
