# Peer review of "Synthesis of Poly-Sodium-Acrylate (PSA)-Coated Magnetic Nanoparticles for Use in Forward Osmosis Draw Solutions"

_nanomaterials, 2019, doi:10.3390/nano9091238_

Round 1

Reviewer 1 Report

The manuscript “ Synthesis of poly-sodium-acrylate (PSA) coated  magnetic nanoparticles for use in forward osmosis draw solutions” described the preparation and characterization of magnetic nanoparticles coated with acrylate polymer and the assessment of their behavior as osmosis draw solution.

The paper is quite well written and contains interesting data, but authors should explain the relevance and novelty of the research presented in the article. Many papers in the last decade report the use of magnetic nanoparticles in acrylate hydrogels in forward osmosis. Since the conclusion of the assessment of the obtained PSA coated MNPs is there are inefficient as draw solution in the reported form, the paper seems to be an unfinished work.

Also, there are many points to be improved, as follows:

1) the title is about “PSA coated magnetic nanoparticles”, while in the introduction the authors refer to the systems synthesized as "polymer -magnetic particles complexes". A clear description in the text corresponding with the title is required.

2) why the 4 nm nanoparticles were selected?

3) Line 343 ”The DLS measurements indicate that the nanoparticles are relatively monodisperse after each purification step”

According to the graph in Fig 7b, that sample is not monodisperse, since the distribution is bimodal! After second and third ultrafiltration step followed by sonication is monomodal. A discussion on the intensity mode DLS diagram is most suitable to consider that no aggregation occurs.

However, the polydispersity could be evaluated from the polydispersity index, not from the aspect of the DLS diagram. To support the sentence above mentioned the authors should compare the PdI values of sample after various purification steps.

4) Line 267 The FTIR spectra should be discussed in more details in order to explain the interaction polymer – MNP surface

Marking the significant peaks on figure 4 will ensure better understanding of the text

5) Line 272 “TGA analysis and mass spectroscopy” What mass spectroscopy?

6) “The results and discussion” section should be re-organized for a better view of characteristics of the PSA coated nanoparticles (for example thermogravimetric analysis data discussed in the same place, since there are related to the same issue, the polymer attached to the nanoparticles).

7) line 70 “aqueous solution” not aqua.

8) Line 497 “The PSA-MNPs showed good colloidal stability in aqueous solution” The statement is not supported by experimental data. No stability test is presented.

9) The references are most of them not relevant and quite old, in the last 5 years there are many papers related to the use of magnetic nanoparticles in acrylate hydrogels as draw solutions.

Reviewer 2 Report

The manuscript 563926 deals with a suitable topic for Nanomaterials Journal, since the functionalization of magnetic nanoparticles for specific applications can be considered a field of notable interests for a number of researchers. However, before accepting it for a possible publication, the following moderate/minor revisions should be followed:

1) English grammar should be revised, some sentences are not clear and the authors should correct some typos present in the Whole manuscript;

2) The authors should check the format used for the text as well as for the tables, figures and equations. The equations must be written by using the equation editor, and should not be included as images, then some words have been underlined (see below equation 1), without a proper reason. The format should be completely revised. For instance, when the parameter of an equation were defined, the authors changed the font and this should be avoided and in paragraph 3.2 some words were written in bold;

3) The authors may report with more detail the synthesis procedure. For instance, they should report the mixing intensity (rpm) and final pH of the MNPs suspension;

4) The authors should check carefully the references, since some names have been erroneously reported. Moreover, the font used to write the references is not the same used in the Whole manuscript;

5) Which refractive index have been used the authors for DLS analysis? It should be reported in materials and methods section;

6) The authors may report some comparisons among the Z potential of their particles with those reported in literature. For this reason I suggest some updated works surely present in the Nanomaterials journal background.

Reviewer 3 Report

To dear editors

This paper studied the synthesis of the draw solutes for forward osmosis system. They fabricated magnetic nanoparticles stabilized by poly sodium acrylates which enhance the osmotic pressure. They found the best combination ratio of MNPs/PSA. Although this study would be appropriate for forward osmosis development, you have to make sure some issues.

Q1 : Many stabilizers were used to synthesize nanoparticles. Just, I wonder why you did not use PSA during MNP fabrication? Also, to make stabilized nanoparticle dispersion, the steric and coulomb repulsive forces are very critical factors. Can you say what force is a critical factor in your systems? 

Q2: In Fig. 3, the particle size of MNP is around 4 nm. But, they are already separated, as a stabilizer is used. Is it possible? NO aggregation?  Why did not show the virgin MNP particle size distribution via DLS measurement?

Q3: In forward system, the polymeric contents could be the serious fouling materials to decay the water flux. In Fig. 11, the flux decreases with increasing the operation time. However, when NaCl is used as the draw solution in Fig. 13, the declining trend of water flux is not observed. Why? As you know well, the salt size of Na or Cl is much smaller than PSA. Thus, there is the salt flux (Js). 

There are some typing errors; Please double-check.

#70 : aggregate in aqua solutions ==> in aqueous solutions

#148, 151 : Please remove the underline

#158, 167 171~175, 198 : Please make sure the format of the word

#339: what is the "d" value. You have to describe the definition of d.

## 176 : Characterisation by Thermogravimetric analysis --> small letter

Round 2

Reviewer 1 Report

I agree with the publication of the manuscript in the revised form